# Impact of Flax and Basalt Fibre Reinforcement on Selected Properties of Geopolymer Composites

**Miroslav Frydrych [1], Štěpán Hýsek [2,3,\*], Ludmila Fridrichová [1], Su Le Van [2], Miroslav Herclík [1], Miroslava Pechočiaková [4], Hiep Le Chi [2] and Petr Louda [2]**

1. Department of Textile Evaluation, Faculty of Textile Engineering, Technical University of Liberec, Studentska 2, 461 17 Liberec, Czech Republic; miroslav.frydrych@tul.cz (M.F.); ludmila.fridrichova@tul.cz (L.F.); miroslav.herclik@tul.cz (M.H.)
2. Department of Material Science, Faculty of Mechanical Engineering, Technical University of Liberec, Studentska 2, 461 17 Liberec, Czech Republic; longsuvp90@gmail.com (S.L.V.); lechihieptul09@gmail.com (H.L.C.); petr.louda@tul.cz (P.L.)
3. Department of Wood Processing and Biomaterials, Faculty of Forestry and Wood Sciences, Czech University of Life Sciences in Prague, 16521 Praha, Czech Republic
4. Department of Material Engineering, Faculty of Textile Engineering, Technical University of Liberec, Studentska 2, 461 17 Liberec, Czech Republic; miroslava.pechociakova@tul.cz
* Correspondence: hysekstepan@seznam.cz; Tel.: +420-607-501-858

**Abstract:** The submitted paper deals with the physical and mechanical properties of geopolymer composite materials reinforced with natural fibres. For this study, we aimed to develop a geopolymer composite reinforced with long flax fibres, which were implemented in the geopolymer in the form of a nonwoven fabric that reinforced the structure of the geopolymer over the entire thickness of the board. In order to compare the properties of the developed composite with natural fibres, a geopolymer without fibres and a geopolymer reinforced with basalt fibres were also produced. The monitored mechanical properties were impact bending, bending strength and compressive strength. Differential scanning calorimetry (DSC), thermogravimetric analysis (TGA), Fourier transform infrared spectroscopy (FTIR) and microscopic analysis were also carried out. The results clearly showed the positive effect of the addition of natural fibres on impact bending and bending strength. However, the addition of natural fibres in the form of a nonwoven fabric significantly increased the variability of the properties of the developed composites. In addition, a different pattern of joint failure was noted between geopolymer reinforced with flax fibres and geopolymer reinforced with basalt fibres.

**Keywords:** geopolymer; natural fibre; flax; basalt; reinforcement

## 1. Introduction

Reinforcing polymers with fibres can create high-performance materials. The specific mechanical properties of the materials that use high-quality natural fibres achieved better values than composites reinforced by man-made fibres that had already been developed [1]. Flax is a very strong natural fibre with a tensile strength of 1.5 GPa and a specific tensile strength of GPa·m$^3$·kg$^{-1}$. Thanks to these properties and their renewability, flax fibres are considered an environmentally friendly alternative to glass fibres for use in composites. These fibres have a number of other advantages compared to glass, basalt or carbon fibres: they do not cause skin irritation, their edges blunt less, they do a very good job of absorbing energy, vibration and UV radiation, they do not create static charge, they are resistant to insects and bacteria, they are harmless to health, they are biodegradable and they do not release VOCs.

Combined with low density and high strength, they are designed for material use in composites [2,3]. On the other hand, their disadvantages are degradation at lower temperatures, higher variability of mechanical properties, lower maximum tensile strength, lower relative elongation and low resistance to natural impacts. The free and bound water content also poses problems in the manufacture of composites made from natural fibres [2,4,5].

The higher absorption capacity of natural fibre composites makes them an intuitive choice for use in automobiles, where they are able to absorb a significant amount of impact energy. In addition, composites with these fibres are not fragmented [4,6]. Adversely, the high variability of their properties makes it more difficult to use them in the automotive industry, and it is therefore necessary to seek out other applications. One solution may be to use natural fibres in building composites. In recent years, geopolymer composites have appeared as a progressive material [7,8].

A geopolymer is formed by the alkaline activation of aluminosilicates and consists of a repeating unit of sialate monomer (–Si–O–Al–O–). This material is considered a third-generation cement and has many interesting properties [9]. Geopolymers are environmentally friendly building materials that have excellent fire, strength, thermal insulation and acoustic properties. Another of their advantages is the possibility of foaming, which makes it possible to regulate and optimise their properties [10–12]. Geopolymers can be well reinforced, for example, with basalt fibres. This combination is suitable for high-temperature applications [13].

If a geopolymer mixture is produced by mixing, it is not possible, for technological reasons, to use reinforcing fibres that are longer than 32 mm. Longer fibres tend to wind around the mixing propeller and it is not possible to distribute them by blending throughout the whole mixture. If we want to use natural fibres made from flax, which can be longer than 10 cm [14], it is necessary to choose an appropriate application method. The submitted research describes a method of reinforcing geopolymer with long flax fibres in the form of nonwoven fabric. We assume that the implementation of fibres into the geopolymer in the form of nonwoven fabric will have a synergistic effect, and the resulting composite will achieve high strength characteristics due to the oriented fibre structure. In order to compare the properties of the developed reinforced flax fibre composite, a geopolymer reinforced with basalt fibres and a geopolymer without reinforcement were produced.

## 2. Materials and Methods

### 2.1. Materials

The flax fibres were supplied by Holstein Flachs GmbH (Alte Ziegelei, Germany) and consisted of purified flax fibres. The average fibre length was 5.9 mm and the softness was 67 dtex. Before fibre processing, the fibres were not dried, however they were air conditioned at 20 °C and 65% relative humidity for four weeks and they reached an equilibrium moisture content of 10%. The flax fibres were implemented in the composites in the form of fibre mat. The flax fibre clusters were fed into a carding machine (Figure 1) equipped with a set of rollers with wire working coatings. The fibre web emerging from the carding machine was laminated to form a non-reinforced fibre mat with a base weight of 225 g/m$^2$.

Basalt chopped fibres supplied by Orlimex CZ (Czech Republic, Usti nad Orlici) with a length of 6 mm were used. The fibre diameter was 18 μm. The suitability of basalt fibres for use with concrete is declared by the producer. The basalt fibres were implemented into the geopolymer by mixing, as opposed to flax fibres, which were implemented into the geopolymer in the form of a fibre mat.

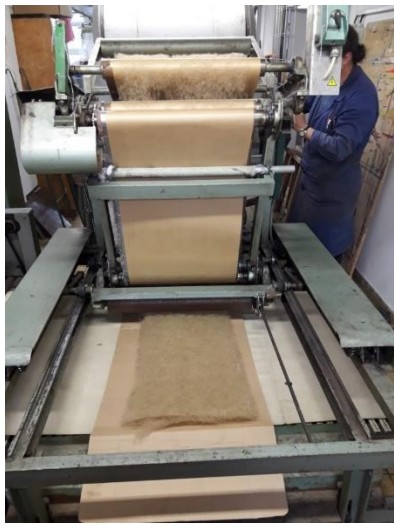

**Figure 1.** Production of a flax fibre mat.

The following manufacturing process was used to produce geopolymer composites. Alkaline Activator Baucis Lk (ČESKÉ LUPKOVÉ ZÁVODY, as, Nové Strašecí Czech Republic) was poured into a Heidolph RZR 2020 rack mixer (Heidolph Instruments GmbH & CO. KG, Schwabach, Germany) and a weighed amount of Baucis Lk metakaolin cement inorganic two-component aluminosilicate binder (ČESKÉ LUPKOVÉ ZÁVODY, as, Nové Strašecí, Czech Republic) was added. The two components were mixed vigorously for 5 min. The mixture was then mixed for 5 min with added fibres (basalt variant only). In the last step, aluminium powder with a purity of at least 99% and an average particle size of 52 μm (PK CHEMIE, Czech Republic) was added, followed by mixing for 30 s. The prepared mixture was poured into moulds with dimensions of 300 × 300 mm. According to the variants, the height of the samples was in the range of 17–20 mm, depending on the interaction of the individual components. The percentage of the individual components used in the production of geopolymers is shown in Table 1. Basalt fibres (6 mm long) were mixed into the geopolymer, and a fibre mat made from flax fibres with a length of 5.9 cm was placed in the prepared mould and the geopolymer mixture was subsequently cast.

**Table 1.** Materials used to produce geopolymers by weight.

| Component | Percentage of Individual Components |
| --- | --- |
| Cement Baucis Lk | 50.45% |
| Activator Baucis Lk | 44.90% |
| Fibres (flax or basalt) | 2.16% |
| Aluminium powder | 2.49% |

*2.2. Methods*

2.2.1. Mechanical Properties

All of the mechanical properties tests were performed after air conditioning the samples at 20 °C and a relative humidity (RH) of 65% for one month. The bending strength test was carried out on the basis of standard ČSN EN 789 [15], where three-point bending was tested and the distance of the supports was 240 mm. The universal testing machine P 100—LabTest II from LaborTech (Opava, Czech Republic) was used, and the test duration was between 45 and 90 s. For this test, specimens with dimensions of 300 × 50 mm and a height of 20 mm were cut from the produced boards. A total of 20 specimens were used. The compressive strength test was carried out on the basis of standard ČSN ISO 1920-10 [16], the universal testing machine P 100—LabTest II from LaborTech was used and the

test duration was between 45 and 90 s. For this test, specimens with dimensions of $50 \times 50$ mm and a height of 20 mm were cut, and 20 pieces were used in total. The impact strength test was carried out on the basis of standard ČSN EN 10045 [17] and the pendulum impact testing machine from Wance was used. For that test, specimens with dimensions of $20 \times 20 \times 150$ mm were cut, and 20 pieces were used in total.

### 2.2.2. Thermogravimetric Analysis (TGA), Differential Scanning Calorimetry (DSC) and Fourier Transform Infrared Spectroscopy (FTIR) Analysis

Samples for TGA, DSC and FTIR analyses were milled and homogenised using a laboratory ball mill and the obtained powder was analysed. A TGA analysis was carried out using the device TGA/SDTA 851 (Mettler Toledo, Greifensee, Switzerland). A temperature program from 25 to 1000 °C was chosen with steps of 10 K/min in the presence of a nitrogen atmosphere with a nitrogen flow rate of 50 mL/min. The DSC analysis was performed using a DSC 3+ device (Mettler Toledo, Greifensee, Switzerland). A temperature program from −50 to 700 °C was chosen with steps of 10 K/min in the presence of a nitrogen atmosphere with a nitrogen flow rate of 50 mL/min. The measuring commenced by conditioning the samples at −50 °C for 10 min. Infrared spectroscopy was performed using a Spectrum One spectrometer (PerkinElmer, Waltham, MA, USA). Obtained powder from composites was placed directly on the attenuated total reflection crystal and pressed with reproducible pressure. The spectral range was recorded from 4000 to 400 $cm^{-1}$ with a count of 10 scans each.

### 2.2.3. Scanning Electron Microscopy

The shape characteristics of the used fibres and the evaluation of composite joint failure were performed using scanning electron microscopy (Tescan Orsay Holding, a.s., Brno, Czech Republic). Ruptured samples from the bending strength tests were used in order to assess the joint failure. The samples of both fibres and composites taken were gold coated using a laboratory coater, and a microscopic analysis was performed using scanning electron microscope MIRA 3 (Tescan Orsay Holding, a.s., Brno, Czech Republic). The following conditions were used: secondary electron detector, acceleration voltage 10 kV, working distance 7 mm.

### 2.2.4. Statistical Data Processing

Descriptive statistics (arithmetic mean, standard deviation) and analysis of variance were used to characterise the obtained data. Tukey's post-hoc test was used to determine if any of the differences between the pairwise means were statistically significant. A significance level of $\alpha = 0.05$ was selected for all of the analyses. The fibre type acts as a factor in the analysis of variance. The impacts of the fibre type on the physical and mechanical properties were shown graphically, and the vertical columns represent 95% confidence intervals.

## 3. Results and Discussion

### 3.1. Characteristics of the Used Fibres

Figure 2a,b shows the surface morphology of the used flax fibres. The figure shows the rough and rugged surface of fibres, but also variability in surface structure. In terms of replacing glass fibres (with a smooth surface and circular cross-section) as reinforcement in the geopolymer with these rough surface fibres, an increase in adhesion between the geopolymer and the reinforcing fibres can be expected, which should lead to higher strength of the resulting composite. This hypothesis will be verified by comparing the geopolymer with flax fibres and with basalt fibres, as shown in Figure 2c.

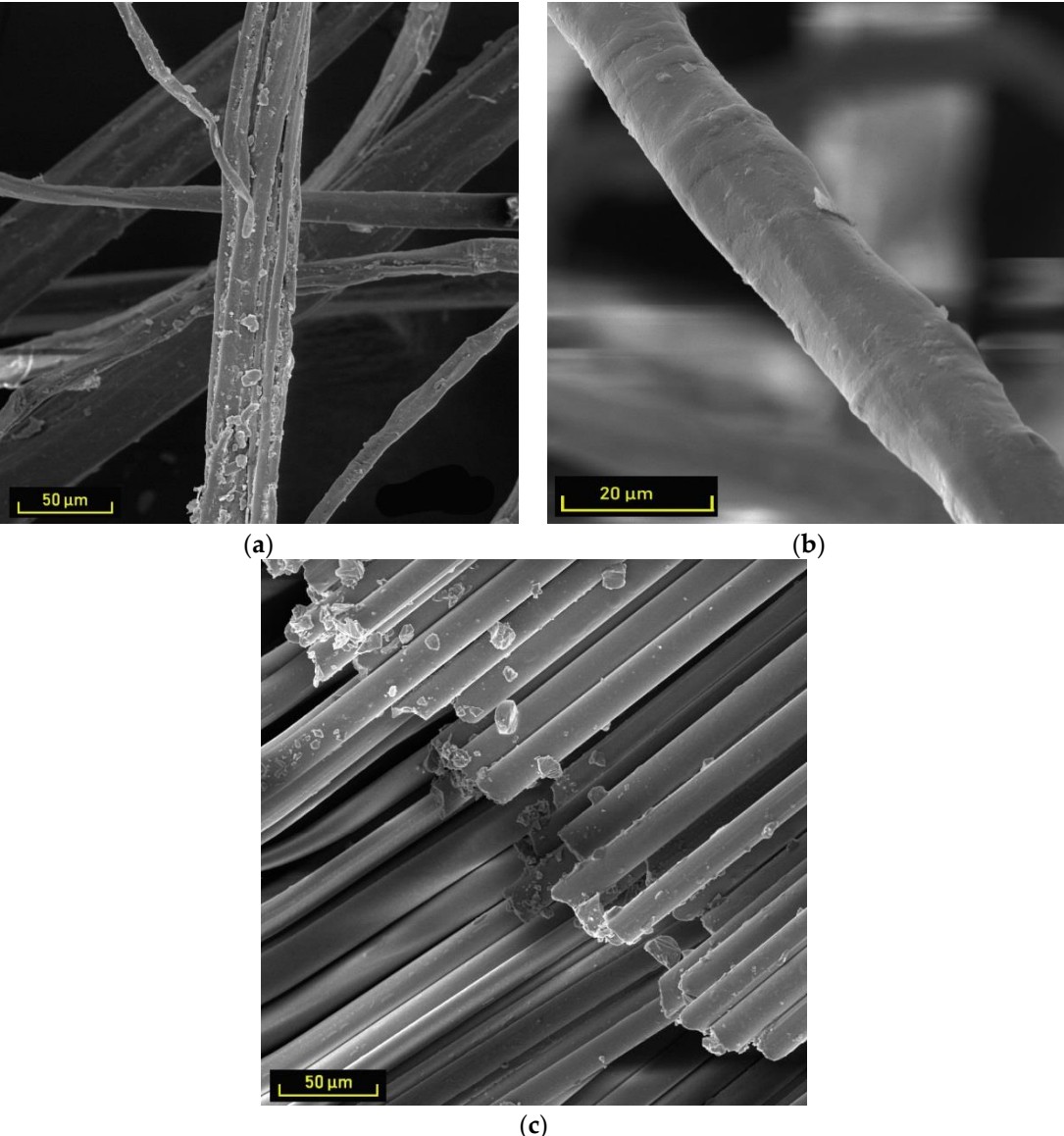

**Figure 2.** SEM figures of used fibres: (**a**) flax fibres, magnification 1000×, (**b**) flax fibre, magnification 4000×, and (**c**) basalt fibres.

*3.2. Impact of the Addition of Fibres on the Mechanical Properties of the Composite*

Table 2 shows the average densities of the produced composites. Due to manual production, there was a higher density variability in samples due to variability in the basis weight of the composites. Flax-reinforced geopolymer composites showed the highest variability due to the higher variability of natural fibres [4] and variability in the basis weight of flax nonwoven fabric [3]. Flax-reinforced geopolymer composites also showed the lowest density due to the highest foaming of geopolymer composites, and therefore the occurrence of larger pores.

**Table 2.** Average densities of composites at 20 °C and relative humidity (RH) 65%.

| No. | Type of Fibres | Density (kg/m$^3$) |
|---|---|---|
| 1 | Flax | 448 (51) |
| 2 | Basalt | 456 (22) |
| 3 | Without reinforcement | 556 (35) |

Values in parentheses are the standard deviations.

Figure 3 shows the effect of added fibres in the geopolymer on impact bending. Samples of the geopolymer reinforced with flax fibre were tested, as well as samples reinforced with basalt fibre, and samples without the addition of fibres were used as a reference set. The graph in the figure shows that the highest impact bending value of 0.62 J/cm$^2$ was achieved by the material reinforced by flax fibres. This difference is statistically significant, at a significance level of 0.05. At the same time however, there was also a high variability of the measured values for this material. This high variability originates both in the high variability of natural fibre properties [4], in the variability of nonwoven fabric properties (caused mainly by the variability of the basis weight) [3], and also in the variability of the basis weight of the resulting composite, caused by worse spillage of geopolymer in the nonwoven textile form. Basalt fibre reinforcement also increased impact bending compared to the reference set, and for samples with added basalt fibres, impact bending was measured at an average of 0.32 J/cm$^2$, and 0.21 J/cm$^2$ for samples without fibres. However, this difference is not statistically significant at a significance level of 0.05. In addition to adding fibres, the impact bending of fibre-reinforced geopolymers can be further increased by increasing the proportion of cement in the geopolymer [18] or by adding a small amount of carbon nanomaterials such as carbon nanotubes or nanofibers [19].

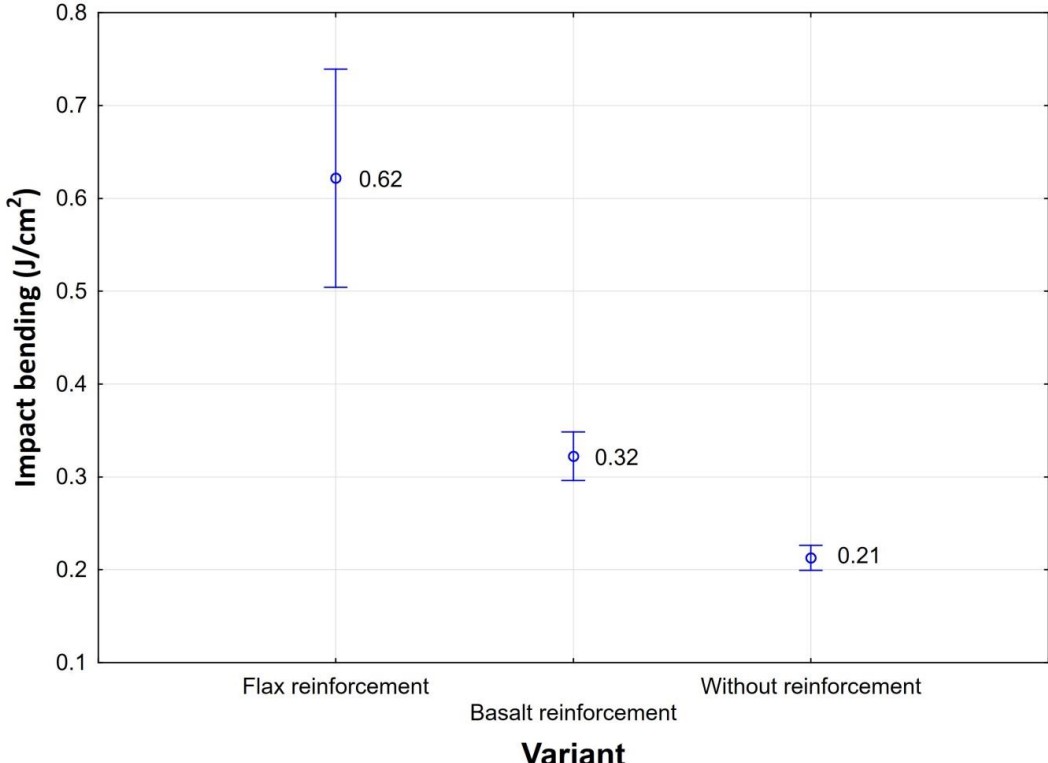

**Figure 3.** Impact of the addition of fibres on impact bending.

Figure 4 shows the effect of fibre reinforcement on flexural strength. It was found that in both cases the flexural strength of geopolymers with added fibres was statistically significantly increased, that is, from 0.59 MPa for samples without added fibres, to 0.91 MPa for samples with basalt fibres, and

to 0.95 MPa for samples with flax fibres. Unlike the dynamic test, this static test did not demonstrate the effect of the fibre type on flexural strength. This finding corresponds to the literature [2,5] where natural fibres are reported to have a higher absorption capacity than artificial fibres. Overall, the bending strength values are set lower than in similar research on geopolymers reinforced with flax fibres [18], but in this research, the developed composites have a density several times higher. The resultant properties of impact bending and bending strength could be significantly improved by using a coupling agent or functionalising the fibre surfaces [20], however, the goal of this study was not to maximise the properties of composites.

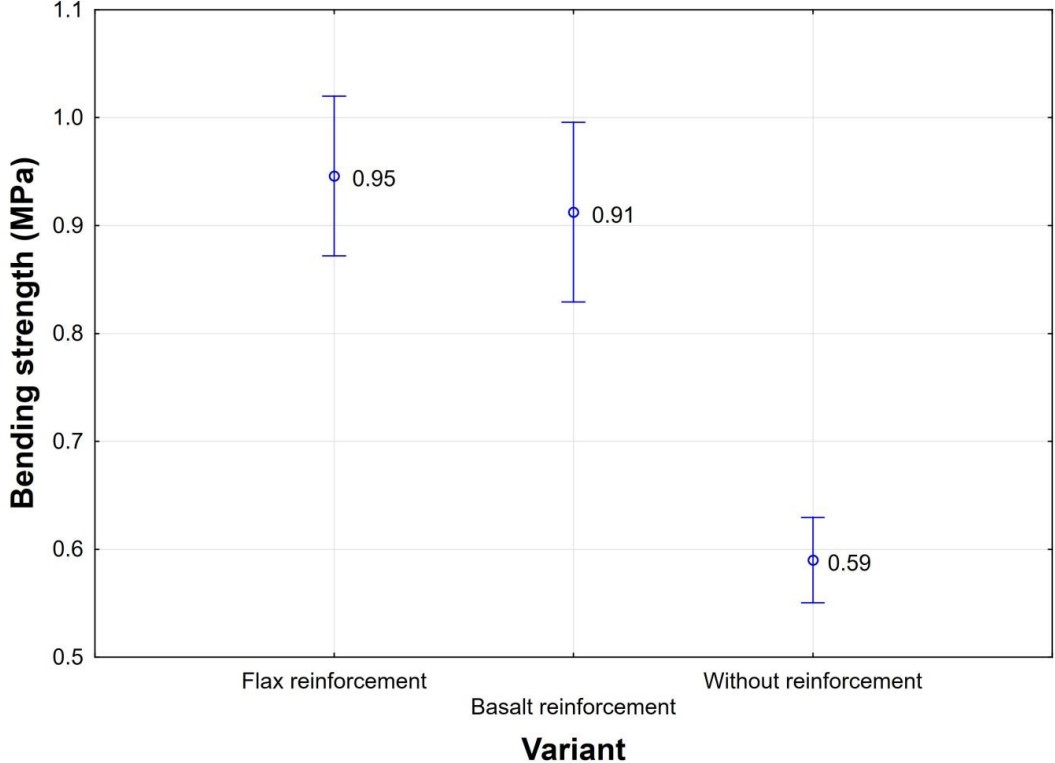

**Figure 4.** Impact of the addition of fibres on bending strength.

The durability of natural fibres as reinforcements in geopolymer composites is affected by the alkalinity of the activators of geopolymer matrices. The alkaline environment is the main reason for natural fibre degradation in cementitious matrices [21]. The results of the tests showed that the mechanical properties (either static or dynamic) of the geopolymer were significantly improved by the addition of flax fibres. Thus, the activator used in the form of an aluminium powder caused sufficiently rapid curing of the geopolymer, and the flax fibres were not decomposed by the alkaline mixture. Further reduction of the curing time of the geopolymer can be achieved, for example, by using nanoclay [22] or nanosilica [23].

The results of the compressive strength test are shown in Figure 5. The graph shows that by adding fibres to the geopolymer, the composite properties are not improved in terms of compressive strength. The highest compressive strength of 0.62 MPa was measured for samples without added fibres. The lowest compressive strength value of 0.33 MPa was measured in samples reinforced with flax fibres. The difference between the compressive strength in samples without fibres and samples with basalt fibres is not statistically significant; the decrease of compressive strength in the samples with flax fibres is statistically significant, at a significance level of 0.05. The low compressive strength of samples with flax fibres is given by the low density of these samples (Table 2), or by the higher number of pores in the geopolymer. The bending strength and the impact bending of the flax fibre

composite was improved, however, the compressive strength decreased significantly. Based on these results, it could be pointed out that the fibres exert a toughening effect on the material, acting as an impact modifier.

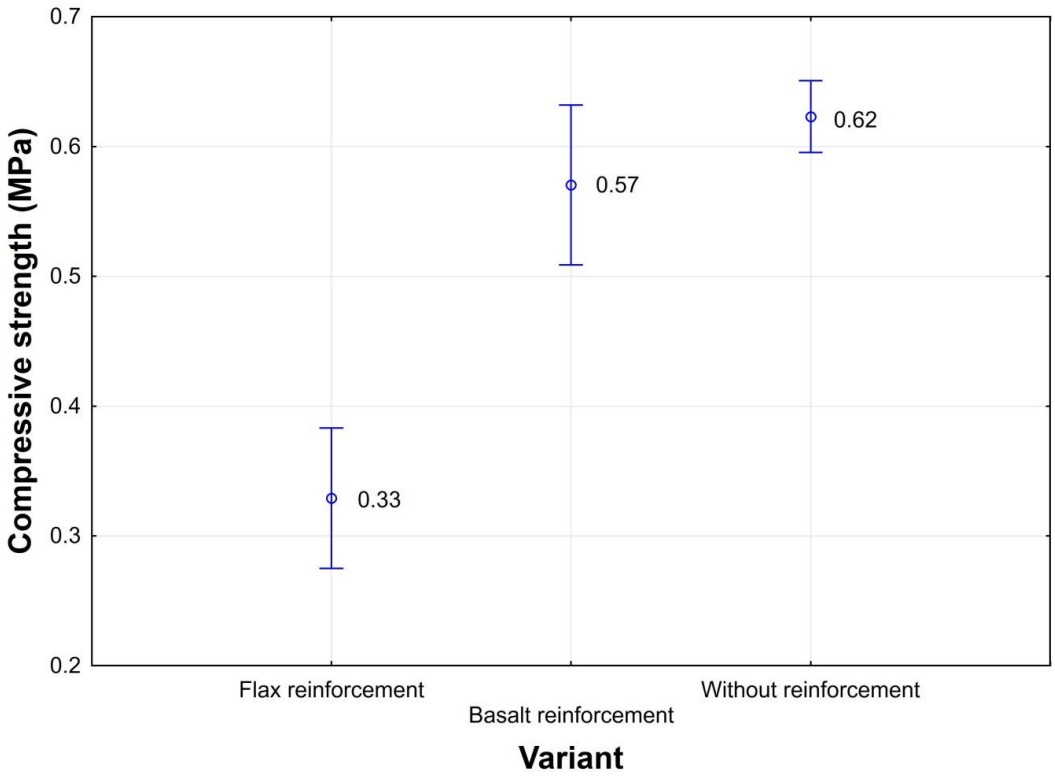

**Figure 5.** Impact of the addition of fibres on compressive strength.

*3.3. Characteristics of Joint Failure*

Electron microscopy showed a different pattern of joint failure for composites with flax and basalt and without fibres. Both flax and basalt fibres exhibited reinforcing properties in the geopolymer, and during bending they were able to transfer the tensile load. Because of the rough surface, flax fibres were better anchored in the geopolymer matrix than basalt fibres, which have a smooth surface. Basalt fibres have higher tensile strength, but due to their smooth surface, they are not able to transfer the maximum possible tensile stress. The mechanisms of failure for flax and basalt fibre composites are therefore different. Figure 6a shows the failure of the geopolymer with flax fibre—breakage of the flax fibre that is fully anchored in the geopolymer matrix. This failure mode in these types of composite materials corresponds to the literature [24]. In terms of basalt fibres (Figure 6b), the fibres do not break, but rather peel off from the geopolymer. This is due not only to the high tensile strength of basalt fibres [25,26], but also to their smooth surface. In Figure 6c, only geopolymer breakage is visible; the fibres were not present in the reference set.

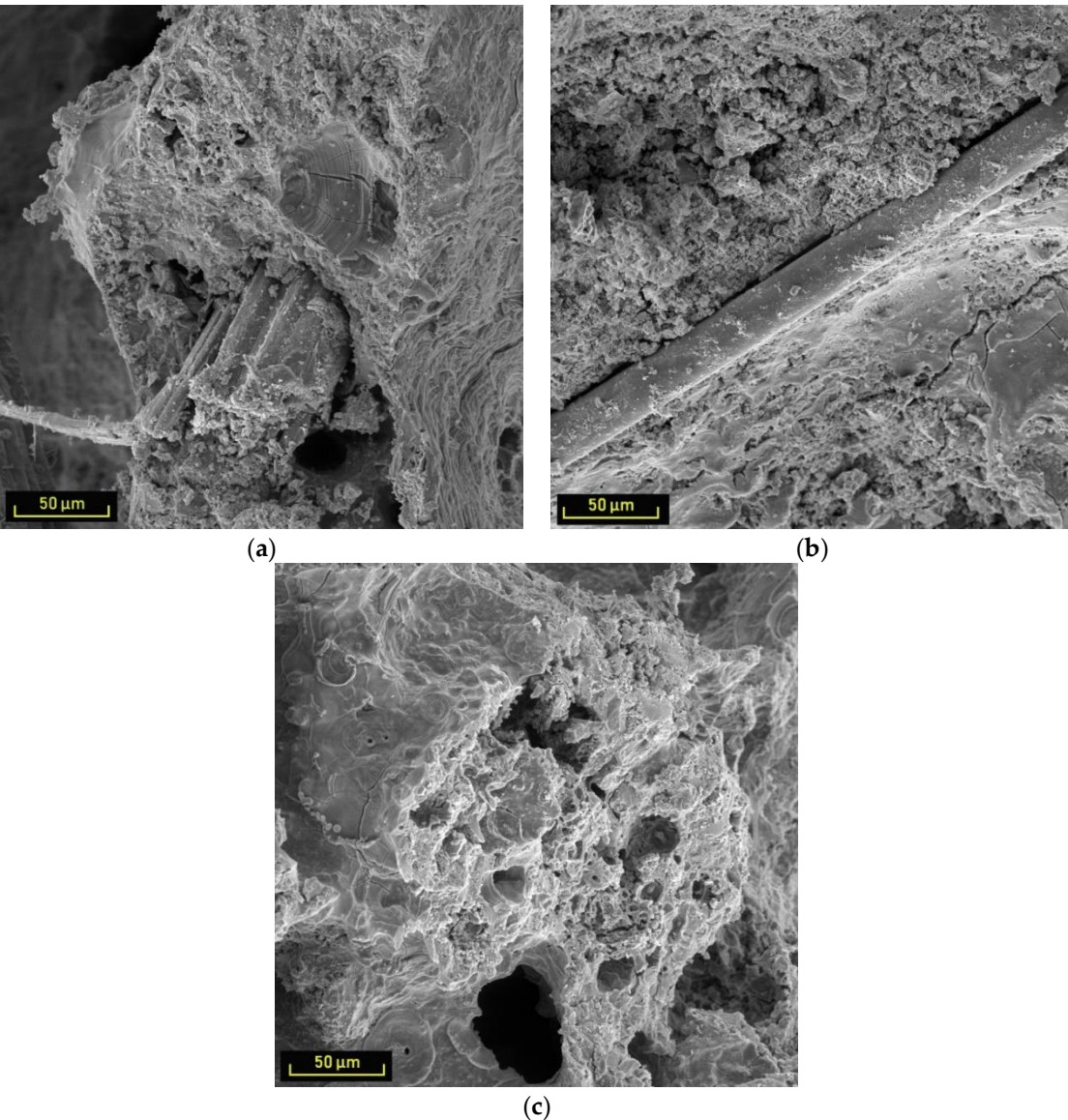

**Figure 6.** Impact of the addition of fibres on the nature of the joint failure (SEM figures): (**a**) flax fibre-reinforced geopolymer, (**b**) basalt fibre-reinforced geopolymer, and (**c**) geopolymer without fibres.

### 3.4. TGA, DSC and FTIR Analysis

Figure 7 captures the measured curves from the TGA analysis. Cellulose degradation is apparent at a temperature of 260.74 °C in the geopolymer filled with flax. This does not occur in the case of the two remaining composites. For all three types of geopolymers, over a temperature range of about 690 to about 700 °C, there is evident melting of the aluminium that was added to the geopolymers in order to activate the foaming reaction. The analysis shows that all three types of composite materials are thermally stable and the captured curves contain no significant jumps. The highest weight loss of the sample is manifested in the temperature range from 25 to 200 °C, which is caused by a loss of moisture (both loose and bound) due to heating. The individual relative decreases are 4.3% for geopolymer without fibres, 4% for geopolymer with flax fibres and 5.7% for geopolymer with basalt fibres. Most of the water in the geopolymers evaporates to 100 °C, whereas the remaining hydrated water leaves at the interval from 100 to 200 °C. Further weight loss is above 600 °C, probably due to sialate bonds and the release of hydroxyl ions [27].

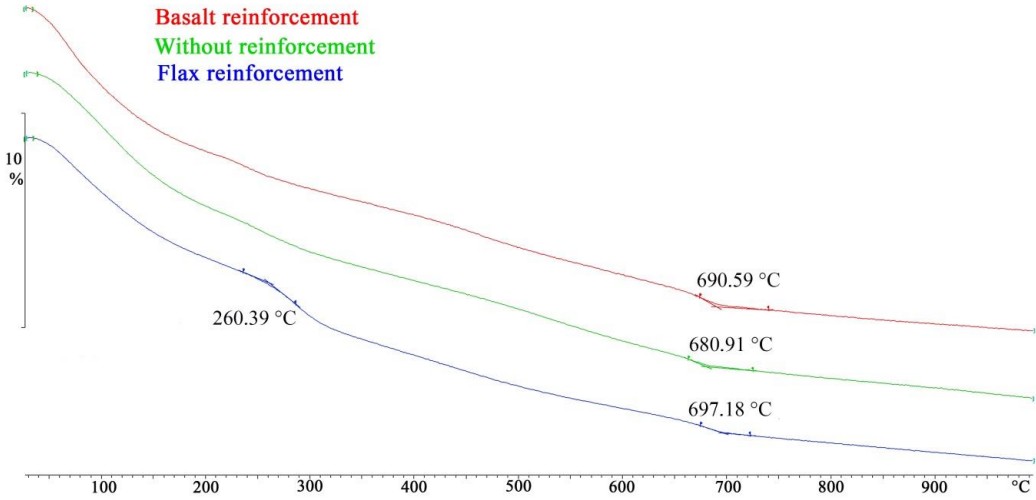

**Figure 7.** Thermogravimetric analysis (TGA) of geopolymer samples.

Figure 8 captures measured curves from the DSC analysis, and these results correspond to the performed TGA analysis. All three types of composite materials show a significant decrease in water content at temperatures ranging from 60 to approximately 90 °C. Endothermic peaks in the range of 250 to 290 °C can be caused by the melting of geopolymer components or by the breakdown of their bonds. IR spectroscopy could be used for their precise specification. In the case of fibre-reinforced composites (flax and basalt), a slight endothermic reaction is evident at 374 °C for geopolymer with flax and at 376 °C for geopolymer with basalt. This reaction can be attributed to degradation of the bonds between the fibres and the geopolymer. At the end of the observed temperature range, the melting range of aluminium is again apparent for all three types of geopolymers.

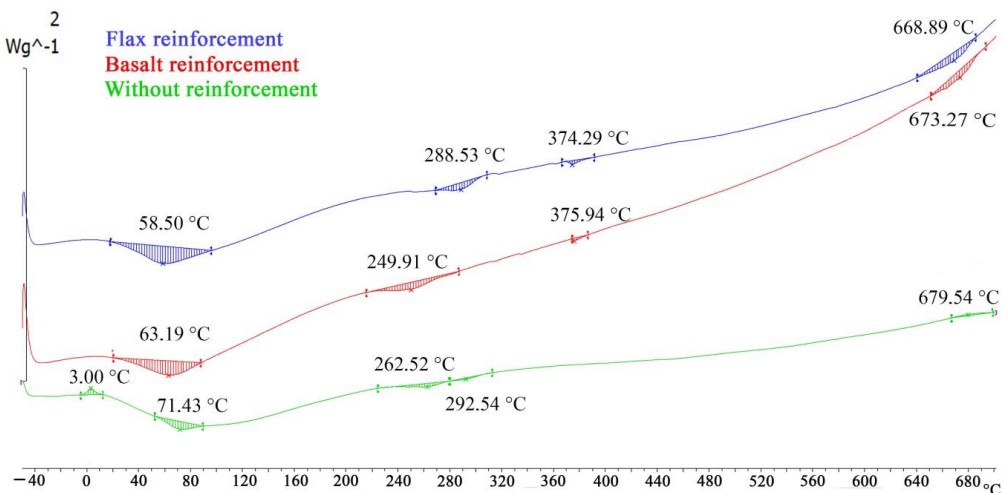

**Figure 8.** Differential scanning calorimetry (DSC) of geopolymer samples.

Figure 9 captures measured spectra from the FTIR analysis. From the figure can be seen that the obtained signals exhibited only slight differences between the variants. Very few differences are due to the absolutely prevailing inorganic matrix. The weight ratio of the fibres used was only 2.16%. The differences caused by the types of reinforcing fibres used can be seen at the wavelengths of 1430–1460 cm$^{-1}$, which correspond to C–H deformation (asymmetric) and aromatic skeletal vibration [28]. The high peak around 1000 cm$^{-1}$ represents vibration Si–O from silicate, and vibration C–O from cellulose and skeletal vibration. From the performed chemical analyses, it can be concluded that the reinforced inorganic matrix is stable and is not negatively affected by the reinforcing fibres.

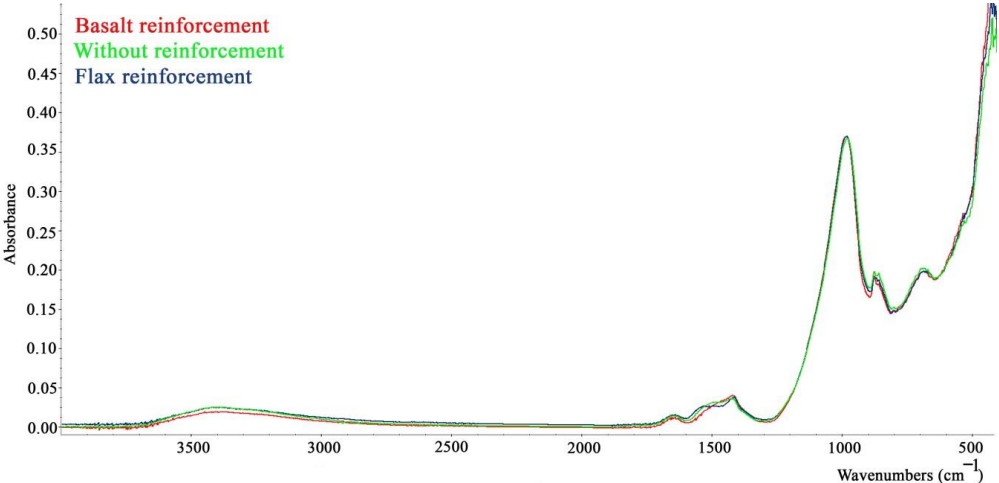

**Figure 9.** Fourier transform infrared spectroscopy (FTIR) of geopolymer samples.

## 4. Conclusions

In the present research, the geopolymer was reinforced with flax fibres, with the flax fibres being implemented in the geopolymer as a nonwoven fabric and the fibres spread over the entire board thickness. The following conclusions can be drawn from the results. Flax reinforcement of geopolymer composites with a density of 448 kg/m$^3$ increased the impact bending of composites up to 0.62 J/cm$^2$; however, the variability also increased. While impact bending of flax-reinforced geopolymers was statistically significantly higher than basalt-reinforced geopolymers, the bending strength of both composites reinforced with fibres was comparable. Geopolymers reinforced with basalt exhibited a different pattern of joint failure than geopolymers reinforced with flax. While basalt fibres peeled off when the composite was breached, flax fibres remained anchored in the geopolymer and ruptured.

**Author Contributions:** Conceptualization, L.F. and P.L.; Data curation, M.F. and M.P.; Investigation, Š.H., S.L.V., M.H., M.P. and H.L.C.; Methodology, Š.H.; Project administration, M.F. and P.L.; Writing—original draft, M.F. and Š.H.; Writing—review and editing, L.F. and P.L. All authors have read and agreed to the published version of the manuscript.

**Funding:** This research was funded by the Ministry of the Interior of the Czech Republic, grant number VI20172019055. The APC was funded by the Technical University of Liberec.

**Acknowledgments:** The results of the project "Application of Geopolymer Composites as Fire Barrier, AGK", registration number VI20172019055, were obtained through the financial support of the Ministry of the Interior in the program "The Safety Research of the Czech Republic" 2015–2020 (BV III/1-VS). This research was financed in part through a grant provided by the Technical University of Liberec—project SGS no. 21302.

**Conflicts of Interest:** The authors declare no conflict of interest.

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
