# Peer review of "Impact of Flax and Basalt Fibre Reinforcement on Selected Properties of Geopolymer Composites"

_sustainability, doi:10.3390/su12010118_

Round 1

Reviewer 1 Report

The research work of Frydrych et al describes the development and characterization of geopolymer composites toughened with natural fibers of flax and basalt. The study has novelty and can be of interest for materials scientists in the area of inorganics. I suggest to improve the manuscript based on the following the points:

-Please explain in the Introduction the concept of "geopolymer" and summarize their manufacturing methods

-Remove Table 3, 4 and 5 and merge data with their respective Figures

-Explain in detail the possible mechanism of failure of each composite

-Have authors consider the possibility of using a coupling agent or functionalize the fiber surfaces? The resultant properties could be significantly improved

-Results show that impact properties are increasing while mechanical strength is decreasing. Therefore, the word "reinforcement" is not defying the actual effect of the fibers on the ceramic. Authors should consider to change the title and revise the discussion accordingly.

-Describe the thermal degradation process of the geopolymer and discuss the catalytic effect of the fibers, which is clearly observable in the DSC curves.

-Did the authors dry properly the fibers? This process is critical for fiber-based composites and the water content of cellulosic fibers can be as high as 10-20wt%. Authors should indicate the water contents of these fillers and if any drying process was performed.

English style must be improved

Author Response

Dear reviewer,

thank you very much for your valuable suggestions. Please find enclosed to this cover letter the revised manuscript and responses to your comments.

I believe that after carefully performed revisions, the manuscript deserves to be considered for publication in the journal Sustainability.

Reviewer 2 Report

Dear author. You have done a good work but I have a few little commentaries.

In table 1 please change 50.45 % with 50.45%. And also for the other cells in the table.

Please correct figure 2.

In figure 2 even you have a rough surface you face also many impurities which doesn`t improve the adhesion without applying a washing process. Even the basalt fibers have many impurities. The surface of the basalt fibers is like the surface of glass fiber.

Please enhance the explanation in 3.2. Impact of the addition of fibres on the mechanical properties of the composite and 3.4. TGA and DSC analysis.

In 3.3. - Even the interface does not break it doesn`t mean that the composite have better qualities. Maybe a mixture of two.

Author Response

(The authors gave the same response as above.)

Reviewer 3 Report

This paper studies the effect of flex and basalt fibers on the mechanical properties of a geopolymer mixture. Flexural testing was conducted to evaluate the effect of the fibers on the flexural strengths. The compressive strength and some other properties are reported. While the topic is interesting, the paper has some major problems that must be addressed before it can be accepted. 

Here are some detailed comments and suggestions:

(1) The technical contribution is unclear. There are many papers on the effect of fiber reinforcement on the mechanical properties of geopolymer. 

(2) A lot of important information of the experimental program is missing. I did not find accurate description of the specimens, number of specimens, specimen preparation, test set-up, loading method, etc. Also, the descriptions of experimental results should be improved. The caption of figure 2, indicates multiple figures designated a to c. You need to specify which figure is a, which one is b, and which one is c. Please refer to the following papers to improve: Meng, W., & Khayat, K. H. (2016). Mechanical properties of ultra-high-performance concrete enhanced with graphite nanoplatelets and carbon nanofibers. Composites Part B: Engineering107, 113-122. Xu et al. (2019). Multiscale investigation of tensile properties of a TiO2-doped Engineered Cementitious Composite. Construction and Building Materials209, 485-491.

(3) The research results are insufficient for a full-length paper. More significant results or analysis should be added. 

Author Response

(The authors gave the same response as above.)

Round 2

Reviewer 1 Report

The authors have corrected some of the issues but they should also still address the following ones:

1) Remove Table 3, 4 and 5 and merge data with their respective Figures
We would prefer not to delete the tables 3, 4 and 5 because the values in the tables provide information about statistical significance of the differences between arithmetic means. The
graphics are listed only for illustration of the descriptive statistics.

The use of three tables to add an statistical analysis in not justifiable in a scientific article. Please consider to find a way to present these data in the graphs or text.

2)Have authors consider the possibility of using a coupling agent or functionalize the fiber surfaces? The resultant properties could be significantly improved.

We agree that by fibre pre-treatment or using some coupling agent the resultant properties would be significantly improved. However the goal of this study was not to maximize the
properties. As is mentioned in the last paragraph of the Introduction, the goal of this study is to develop a method of implementation of long flax fibres into geopolymer matrix.

Please then explain it better in the text.

3) Results show that impact properties are increasing while mechanical strength is decreasing. Therefore, the word "reinforcement" is not defying the actual effect of the fibers on the ceramic. Authors should consider to change the title and revise the discussion accordingly.

The bending strength as well as impact bending was improved by the fibre reinforcement. As
is mentioned in the discussion, the bending properties were enhanced because fibres are able
to transfer the tensile loading. The compressive strength was decreased, because compressive
strength is mainly affected by density of geopolymer and the density of flax and basalt fibre
reinforced geopolymers was lower. We assume that these results correspond with previously
published researches and are fully justifiable. We would be glad if the reviewer suggests
better word than “reinforcement”, however we assume that this word is commonly used when
fibres are mixed with the composite matrix.

The fibers are exerting a "toughening effect" on the material or they act as a "impact modifier".

4) Did the authors dry properly the fibers? This process is critical for fiber-based composites and the water content of cellulosic fibers can be as high as 10-20wt%. Authors should
indicate the water contents of these fillers and if any drying process was performed.

The information was added to the methodology part. The fibres were air conditioned at 20 °C
and 65% relative humidity for 4 weeks and then the fibres reached equilibrium moisture
content 10%.

This is not a drying process but conditioning. Please amend the text accordingly.

Author Response

Dear reviewer,

thank you very much for your valuable suggestions. New analysis was performed and added to the manuscript Please find enclosed to this cover letter the revised manuscript and responses to your comments.

I believe that after carefully performed revisions, the manuscript deserves to be considered for publication in the journal Sustainability.

Best regards

Reviewer 2 Report

Please corect  figure 2 (like figure 6)

Author Response

Dear reviewer,

thank you very much for your positive review. New analysis was performed and added to the manuscript Please find enclosed to this cover letter the revised manuscript and responses to your comments.

Please correct  figure 2 (like figure 6)

The figure was corrected.

Best regards

Reviewer 3 Report

The quality of the paper is improved. But, it still has many problems. 

(1) The technical contribution of this study is still unclear. According to the current version, this study does not have sufficient technical contribution. However, I think perhaps the authors can revise and provide a strong statement of the contribution. 

(2) There are a lot of grammar errors and ambiguous sentences throughout the paper. The authors are suggested to ask a professional editor to polish the writing. 

(3) The quality of figures should be improved. Some figures are blur.

(4) Figure 6 does not provide important, useful information. It should be removed. 

(5) In Introduction, why is the fiber length limited to 32 mm?  

(6) What are the tested specimens? How many did you test? How did you test?

(7) The paper does not have sufficient, significant findings. The authors may consider to add more important contents. 

Author Response

Dear reviewer,

thank you very much for your valuable suggestions. Based on your suggestions, new analysis was performed. Please find enclosed to this cover letter the revised manuscript and responses to your comments.

I believe that after carefully performed revisions, the manuscript deserves to be considered for publication in the journal Sustainability.

Best regards

Round 3

Reviewer 3 Report

The quality of the paper has been further improved. I feel comfortable to recommend it for publication.